# Interplay between children's cognitive profiles and within-school social interactions is nuanced and differs across ages
Eudald Correig-Fraga [1,2] ✉, Roger Guimerà [1,3] & Marta Sales-Pardo [1,3]

Studies investigating the link between school achievement and social networks have shown that both cognitive and non-cognitive factors are integral to academic success. However, these investigations have predominantly been confined by two limitations: 1) they rarely combine cognitive and social data from the same individuals, and 2) when incorporating social data, it is often unidimensional, focusing only on a single type of relationship among children, such as friendship networks or time spent together. This research builds on prior findings by considering cognitive and social data, including preferences for schoolwork relations, leisure/play relations, and friendships, of nearly 5,000 students from Catalonia (Spain) aged 6 through 15. Our findings indicate that children prefer to interact with those who exhibit similar cognitive profiles, but that their preferences diverge between schoolwork and play-related relations during both primary and secondary school. The diverging preferences of children of older ages suggest a greater understanding of the different purposes and expectations of various social interactions.

School achievement is central to child development due to its impact on various outcomes, including level of education, adult income, and even health and longevity[1–4]. Extensive research has analyzed the factors influencing school achievement, particularly regarding academic skills and cognitive abilities[5–7]; the mathematical modeling of these skills and abilities in children has been a topic of interest among researchers for a long time. The main existing theoretical frameworks accept that cognitive abilities can be grouped into distinct clusters, which include verbal, fluid, and memory abilities, among others[8,9]. Gaining further understanding of how cognitive abilities cluster in children provides insight into the underlying mechanisms of cognitive development.

Parallel to the development of cognitive abilities, children's social networks have also been shown to be related to children's well-being, and their emotional and social development[10–13]. Furthermore, the study of social relationships in school classrooms through the analysis of sociograms has proven useful to identify social status within classrooms[10], improve learning experiences[14], and guide reforming initiatives[15]. These studies typically construct sociograms based on interactions between children, but do not distinguish between the different types of interactions that can arise in educational environments, such as friendship or working interactions.

Indeed, the analysis of multiple interaction types is particularly pertinent in social settings, in which different types of interactions can lead to large differences in sociogram structure[16]. Recent developments in multilayer network analysis provide the necessary conceptual framework to allow for the exploration of multiple types of relationships concurrently[17,18]. These tools thus offer the opportunity to cover a gap in the literature to elucidate the social and cognitive dynamics within classrooms by incorporating different types of interactions. The relationship between school social interactions and academic achievement has also been studied for decades, showing that both social ties have an impact on academic achievement[19–24] and academic achievement can impact children's social relationships[24–27], and that the feedback between the two factors can reinforce success and failure[24,28–30]. This feedback becomes particularly relevant during adolescence, when popularity hierarchies emerge in social relationships and can strongly influence peer interactions and academic outcomes[31,32].

The timing of these feedback effects between academic achievement and social relationships may be influenced by crucial developmental transitions in how children understand and evaluate themselves and others. Indeed, some studies have shown that children's conceptions of intellectual competence undergo significant changes during early elementary school

[1]Department of Chemical Engineering, Universitat Rovira i Virgili, 43007 Tarragona, Catalonia. [2]Innovamat Education S.L., 08174 Sant Cugat del Vallès, Catalonia. [3]ICREA, 08010 Barcelona, Catalonia. ✉e-mail: eudald.correig@urv.cat

years[33,34]. Young children tend to have a global, undifferentiated view of ability that encompasses social behavior, work habits, and academic achievement[33]. At higher grades, this broad conception gradually becomes more specific and, in particular, focused on academic achievement. Simultaneously, children learn to meaningfully incorporate social comparison information in their self-evaluations and peer assessments. Importantly, before age 7, children's self-evaluations and social choices are largely unaffected by comparison with peers' achievement[34], suggesting that the mechanisms linking academic achievement and social relationships may not be fully operational until this developmental milestone.

Here, we study in depth the relationship between the cognitive abilities of children and their social interactions, as well as its evolution during school life. We do so by analyzing a large dataset consisting of cognitive and school abilities achievement measures, alongside sociogram-based academic and leisure (positive and negative) preference data, for nearly 5000 students aged 6 through 15 in 13 Catalan schools (Fig. 1). We surmise that one of the reasons why the progress in understanding the relationship between social ties and cognitive development in children has stalled is the predominant use of partial proxies of social relationships, such as the time spent together by two children. We argue that social relationships between children at school can have different origins; for instance, children might spend time together because they need to work on a project or because they want to play together. Thus, we take a multilayer network approach. We conjecture that examining children's preferences for both academic and leisure activities, both positive and negative, can fill some gaps left by current approaches. Additionally, we investigate in detail the evolution of the interplay between social networks and cognitive profiles along children's age.

## Methods

### Data description

We gathered data from two tests: a psychometric test of school achievement and underlying cognitive abilities and a sociogram. These tests and the sociograms were gathered as part of the schools' participation in a program for the early detection of learning issues. Parents gave informed consent for the use of these data for screening for learning difficulties or special learning needs. The data were not gathered for the purpose of this study and therefore, it can be considered secondary use.

**Cognitive evaluation.** Each participant completed an hour-long computerized test, adapted according to their age. The test included several tasks designed to measure various cognitive aspects and academic achievement elements (task screenshots available in Supplementary Section 1):

- **Execution speed**: A 30-s task where children clicked on a moving target in a $4 \times 5$ grid, measuring motor response speed.
- **Reading fluency**: A 3-min task requiring children to complete sentences by selecting the correct word from five options.
- **Working memory**: Children memorized images, performed intervening visual discrimination tasks, then identified the memorized images from a grid of 24 options.
- **Visual processing speed**: A 2-min task where children identified whether a target symbol appeared among a set of options.
- **Arithmetic fluency**: A 2-min assessment of rapid addition and subtraction within the 0–20 range using a specialized numerical entry interface.
- **Long-term memory**: After a 10-min delay, children recalled images from the working memory task.

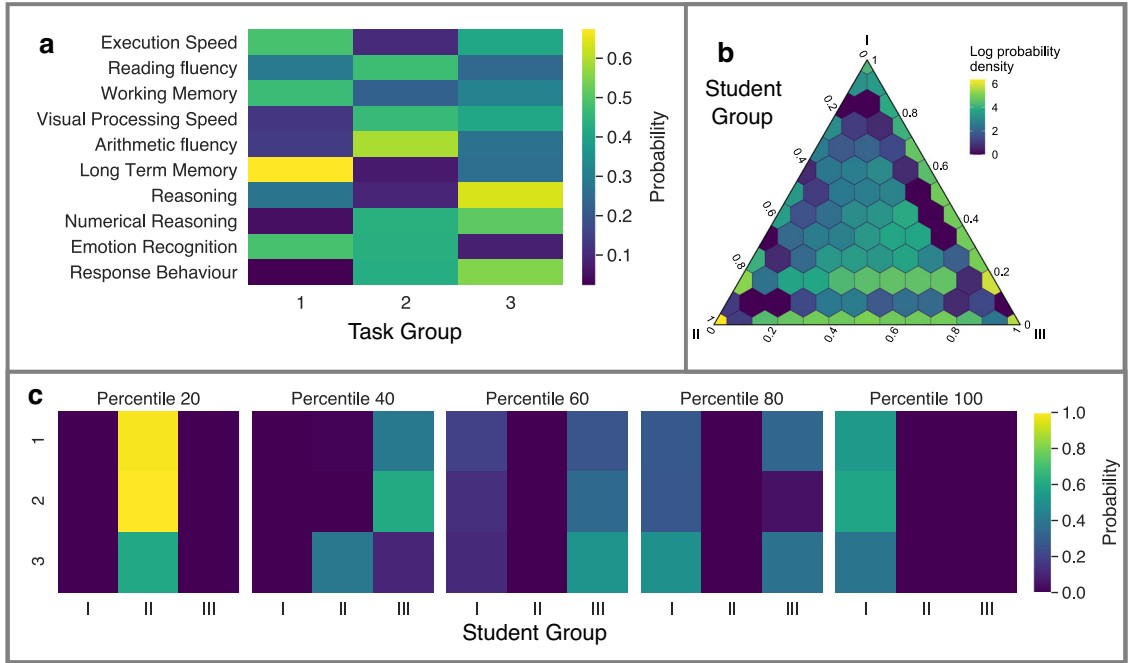

**Fig. 1 | MMSBM results on the cognitive tasks and comparison with other models.** Panels **a**–**c** show MMSBM parameters for the data on the achievement of cognitive tasks. **a** Cognition task profiles. For each task (row in the matrix), we obtain a membership vector (row). Each vector element indicates the probability that a task belongs to task group 1, 2, or 3. Each element is color-coded following the color bar on the right. **b** Student profiles. Ternary plot showing the distribution of student membership profiles. Each point in the plot represents a possible combination of memberships to groups I, II, and III, with vertices representing memberships in only one group. Elements along the side connecting vertices $X$ and $Y$ correspond to mixtures of groups $X$ and $Y$. Elements inside the triangle correspond to mixtures of the three groups, with the barycenter corresponding to a uniform mixture of the three groups. Color represents the fraction of students with that membership composition in a logarithmic scale, according to the color bar on the right. **c** Achievement probability matrices. For each achievement quintile $Q$, we show the matrix of probabilities $P_{XY}(Q)$ that a student in group $X = I, II, III$ has achievement $Q$ in a task in the group $Y = 1, 2, 3$. Matrix elements are colored according to the color bar on the right.

- **Reasoning**: A 3-min task requiring children to identify the next element in visual patterns from four options.
- **Numerical reasoning**: For children above age 9, a 3-min task requiring completion of mathematical equations with 1–3 missing elements.
- **Emotion recognition**: Children identified emotions (happiness, fear, rage, surprise, sadness, or no emotion) displayed in facial expressions.

An additional constructed dimension, **Response behavior**, accounted for the number of errors made throughout the test and aimed to gauge impulsivity in children. Performance on each task was converted to percentiles based on the full sample distribution, from which we extracted quintiles and assigned a score $R$ ranging from 1 to 5. We refer to this complete set of assessments as "cognitive tasks" for simplicity.

**Sociometric evaluations**. The sociogram questionnaires were administered at the classroom level by psychology professionals in a 30-min session conducted in Catalan. While the questionnaire included both indirect and direct questions (see Supplementary Section 2 for full questionnaire details), for this study, we focused on five direct questions where each child named up to three classmates:

- "Who would you like to join your group to do an assignment?"
- "Who would you NOT like to join your group to do an assignment?"
- "Who would you like to join your group to play in the playground?"
- "Who would you NOT like to join your group to play in the playground?"
- "Who are you more friends with?"

Prior to the assessment, students reported their gender, mother tongue, and current state of mind. Responses were recorded as directed links between children (source child $\rightarrow$ nominated peer) and were analyzed in their original form. Both cognitive and sociometric assessments were conducted in the same session.

**Participants**. Over the span of 2 years, one or both of these tests were administered over 8000 times to more than 7000 students aged 6 to 15 in 22 schools in Catalonia, Spain. From these, 4979 students were administered both types of tests (5800 simultaneous tests in total, some students were administered the test twice in separate years). In this study, we analyze the data collected in all these tests, since no data was excluded. In particular, we studied 5057 students, where 46% self-reported as boys, 40% as girls, 13% chose not to answer, and <1% answered "other" gender.

**Mixed membership stochastic block models for cognitive data**
For the modeling of cognitive data, we use a network approach by representing cognitive data as a bipartite graph. We then use mixed membership stochastic block models (MMSBM)[35,36] to find the latent group structure of tasks and students.

Formally, we have $U$ users and $T$ tasks and a bipartite graph $R = \{(u, t)\}$ of links, where the link $(u, t)$ indicates that a user $u$ has scored $s$ on a task $t$, where $s_{ut} \in S = \{1, 2, 3, 4, 5\}$. There are $K$ groups of users and $L$ groups of tasks, and each user $u$ has a vector $\theta_{uk} \in \mathbb{R}^{\mathbb{K}}$ of membership to group $k$, and each task $t$ has a vector $\eta_t \in \mathbb{R}^{\mathbb{L}}$ of membership to group $t$, where $\Sigma_k \theta_{uk} = \Sigma_l \eta_{tl} = 1$, so that $\theta_{uk}$ is the probability that user $u$ belongs to group $k$ (and analogously for task $t$ and task group $l$. For each pair of groups $k, l, p_{kl}(s)$ is the probability that a user that belongs to group $k$ achieves score $s$ on a task that belongs entirely to group $l$ and satisfies the normalization condition $\sum_s p_{kl}(s) = 1$.

In the MMSBM, the probability that user $u$ achieves score $s_{ut} = s$ in task $t$ is:

$$Pr[s_{ut} = s] = \sum_{k,l} \theta_{uk} \eta_{tl} p_{kl}(s) \tag{1}$$

The likelihood of observing a set of $S^O = \{s_{ut}\}$ scores is then

$$P(S^O|\boldsymbol{\theta}, \boldsymbol{\eta}, \mathbf{p}) = \prod_{s_{ut} \in S^O} \sum_{k,l} \theta_{uk} \eta_{tl} p_{kl}(s_{ut}), \tag{2}$$

where $\boldsymbol{\theta} \equiv \{\theta_u\}$, $\boldsymbol{\eta} \equiv \{\eta_t\}$, and $\mathbf{p}$ is the probability matrix.

We infer the values of the parameters $\hat{\theta}, \hat{\eta}, \hat{\mathbf{p}}$ that maximize the log-likelihood (or the log-posterior if we use uniform priors) using an expectation-maximization approach. Specifically, as in Godoy-Lorite et al.[37] we use a variational trick and introduce a latent distribution $\omega_{ut}(k, l)$ that allows us to convert the logarithm of a sum into the sum of logarithms as follows:

$$\log P(S^O|\theta, \eta, \mathbf{p}) = \sum_{(u,t) \in S^O} \log \sum_{k,l} \theta_{uk} \eta_{tl} p_{kl}(s) \tag{3}$$

$$= \sum_{(u,t) \in S^O} \log \sum_{k,l} \omega_{ut}(k, l) \frac{\theta_{uk} \eta_{tl} p_{kl}(s)}{\omega_{ut}(k, l)} \tag{4}$$

$$\geq \sum_{(u,t) \in S^O} \sum_{k,l} \omega_{ut}(k, l) \log \frac{\theta_{uk} \eta_{tl} p_{kl}(s)}{\omega_{ut}(k, l)}. \tag{5}$$

Note that in the last step we use Jensen's inequality, by which $\log \bar{x} \geq \overline{\log x}$ so that $\log \frac{\overline{\theta_{uk} \eta_{tl} p_{kl}(s)}}{\omega_{ut}(k,l)} \geq \overline{\log \frac{\theta_{uk} \eta_{tl} p_{kl}(s)}{\omega_{ut}(k,l)}}$. The equality is reached when

$$\omega_{ut}(k, l) = \frac{\theta_{uk} \eta_{tl} p_{kl}(s)}{\sum_{k'l'} \theta_{uk'} \eta_{tl'} p_{k'l'}(s_{ut})}. \tag{6}$$

To find updated equations for the parameters that we obtain by taking partial derivatives of the log-likelihood, including Lagrange multipliers for the normalization constraints, which leads to

$$\theta_{uk} = \frac{\sum_{t \in \delta u} \sum_l \omega_{ut}(l, k)}{\sum_{t \in \delta u} \sum_{kl} \omega_{ut}(l, k)} = \frac{\sum_{t \in \delta u} \sum_l \omega_{ut}(l, k)}{d_u} \tag{7}$$

$$\eta_{tl} = \frac{\sum_{u \in \delta t} \sum_l \omega_{ut}(l, k)}{\sum_{u \in \delta t} \sum_{kl} \omega_{ut}(l, k)} = \frac{\sum_{u \in \delta t} \sum_l \omega_{ut}(l, k)}{d_t} \tag{8}$$

$$p_{kl}(s) = \frac{\sum_{u,t \in S^O | s_{ut} = s} \omega_{ut}(k, l)}{\sum_{u,t \in S^O} \omega_{ut}(k, l)} \tag{9}$$

where $d_u$ and $d_t$ are the degrees of the user $u$ in task $t$, respectively.

**Expectation-maximization algorithm**. To find $\hat{\theta}, \hat{\eta}, \hat{\mathbf{p}}$, starting from some initial conditions $\hat{\theta}_0, \hat{\eta}_0, \hat{\mathbf{p}}_0$:

1. Expectation step: Compute auxiliary distributions $\omega_{ut}(k, l)$ using $\hat{\theta}_0, \hat{\eta}_0, \hat{\mathbf{p}}_0$.
2. Maximization step: Update $\hat{\theta}, \hat{\eta}, \hat{\mathbf{p}}$ using equations. (7), (8), (9). Iterate steps 1 and 2 until model parameters converge.

**Model selection**. To determine the optimal cluster numbers, both for task and student groups, we performed a hyperparameter optimization process employing a tree-structured Parzen estimator (TPE) optimization algorithm[38]. In the model, we optimize for accuracy while maintaining the most parsimonious model possible, which results in three groups, both for tasks and students (see a detailed explanation in Supplementary Fig. 3).

**Model comparison**
**Comparison of predictive algorithms**. To assess the predictive power of MMSBMs, we run a missing data predictive task by hiding 40, 60, and 80% (20 different splits for each fraction).

We compared the predictive performance of MMSBMS with other popular prediction algorithms such as predictive mean matching (PMM)[39], classification and regression trees (CART)[40], random forests (RF)[41], Bayesian linear regression (BLR)[42], and lasso regression (LASSO)[43]. To run the comparison algorithms, we used the R package *mice*[44] to impute missing data.

## Sociogram analysis

**Sociogram structure.** To analyze communities in the sociogram links, we used a stochastic block modeling (SBM) approach with fixed node memberships using GraphTool[45]. Specifically, we considered four SBM model families for each classroom combining degree and not-degree corrected SBMS as well as using or not a hierarchical prior on the block structure. For each model type, we performed 20 runs, then selected the optimal model (i.e., the model with the minimum description length or optimal posterior).

**Overlaps across sociograms.** We analyzed the overlap across the different sociograms for the same class using the Jaccard index[46], defined as the ratio of the number of common edges between the two networks to the total number of unique edges in the two networks, A and B:

$$J(A, B) = \frac{|A \cap B|}{|A \cup B|} \tag{10}$$

**Age statistics.** To obtain a measure for each age, we averaged over all classrooms for each age.

## Hypothesis testing

All the comparisons we made were comparing a categorical independent variable (gender, course, etc.) and a continuous dependent variable (membership coefficient, Jaccard index, etc.). We tested all dependent variables for normality using a Shapiro–Wilk test and found that none of them were normal, so we performed two-sided Mann–Whintey $U$-tests for all our hypothesis testing when the independent variable was categorical, and Spearman correlation when it was continuous (given that the dependent variable was always continuous). We used the Hodges–Lehmann estimator (HLe) to assess effect size[47]. Throughout this work, the type I error is set to 0.05.

## Study preregistration and consent

This study was not preregistered. All participants gave informed consent to their participation in the data collection process. However, all data were collected prior to and without connection to the present study. Therefore, the present study only made secondary use of the data previously collected. No additional ethics approval was required by our institution for secondary data analysis.

## Results

### Three cognitive profiles along a single achievement axis are predictive of achievement in cognitive tasks for children of all ages

The first question we address is whether there exist groups of children with well-defined cognitive profiles, such that student achievement depends only on cognitive profile and the type of task. To answer this question, in the spirit of Gerlach and collaborators[48], we represent our data as a bipartite graph in which there are two types of nodes, children (students) and tasks, and the relationship between a child and a task is the achievement quintile of the student in that task (see Data and Methods). We then follow a probabilistic approach based on mixed membership stochastic block models[35] (MMSBM), which have been shown to reliably find clusters in bipartite relational data[36,37]. In the present context, using stochastic block models (SBM) amounts to assuming that there exist underlying groups of children and groups of tasks, such that the probability that a child has a certain degree of achievement in a specific task depends only on the groups to which the

child and the task belong. Furthermore, in their mixed membership variant, stochastic block models account for the possibility that students and tasks belong to mixtures of groups, rather than a single group. As a result, each node in the network (child or task) is characterized by a vector that indicates the level of membership to each of the underlying groups (see Methods for details). In our context, a student's membership vector essentially represents their cognitive profile, and a task's membership vector portrays its cognitive demands.

To show that our probabilistic approach provides better description of our data than other commonly used approaches (Supplementary Fig. S10, and Methods for details), we perform prediction experiments in which we fit the models to a partial observation of the data and measure the predictive performance of the different models on the unobserved data. Using performance in predictive tasks as a criterion for model selection has been shown to be a good proxy for more rigorous approaches[49]. We find that the MMSBM approach is superior to all other techniques we consider, for different amounts of unobserved data and both in terms of predicting the exact achievement level and of being at most one quintile away from the correct answer (one-off prediction) (Supplementary Fig. S10). Therefore, we conclude that the MMSBM provides the most accurate description of our data, and therefore use children and task membership vectors as the best way to define the cognitive profiles of children and the cognitive demands of tasks (see Methods, and Supplementary Fig. S11 for model parameter selection).

We find that the best, most parsimonious description of our data consists of three underlying groups of children and three underlying groups of tasks (Supplementary Fig. S11). Our analyses show that the majority of tasks are mostly associated with a single task group, and that tasks associated with the same group are related to a specific cognitive area (Fig. 1a): attention and cognitive control (group 1); cognitive function and task complexity (group 2); and reasoning and reaction behavior (group 3). The sole exception in our task-group classification is emotion recognition, a task that requires students to select the emotion displayed in a picture of a person making a facial expression. While this task is not an attention-related task, it is classified in group 1. This is because students perform similarly in this task and in attention-related tasks (Supplementary Fig. S11), which can be explained because this is the final task of the hour-long test, so that children with a lower ability to sustain attention for long periods of time typically also performed worse.

In terms of achievement, children groups are very clearly defined across all tasks. Group I is associated to high achievement, group II is associated to low achievement, and group III is associated to average achievement (Fig. 1c). In contrast to tasks, children are not always associated to a single student group. However, their group memberships follow a very structured pattern (Fig. 1b). We find many children who belong almost exclusively to group II (low achievement), but find no children who belong exclusively to groups I or III. Furthermore, we find that children do not mix groups I and II (low and high achievement), but that many students are either split between groups II and III (low and average achievement) or between groups I and III (high and average achievement). In terms of gender, we find equivalent membership patterns for groups II and III, and only a significant, but very small increase of boys in group II (low achievement) group [two-sided Mann–Whitney $U$-test, $p < 0.001$; Hodges–Lehmann estimator $< 0.001$ for group II (low), and $p > 0.05$ for groups I and III] (see Supplementary Fig. S16). This observation is consistent with studies that find small but significant advantages for girls in compulsory education settings (see ref. 50 for a comprehensive meta-analysis).

In cognitive terms, our latent classification of children and tasks supports the idea of a single core dimension of cognitive ability or the so-called positive manifold of cognitive testing[51–53]. Furthermore, we find that this latent structure describes very well the scoring patterns of students, despite individual variability. Indeed, while 35% of the students have scores that are in high discrepancy with those of their assigned group (Supplementary Fig. S15), 92% of these only show high discrepancy in at most two tasks

(Supplementary Fig. S15), showing a good agreement between model expectations and observed data. Therefore, by grouping students along a single achievement axis, our clustering approach reproduces the psychometric construct of the g-factor[54], and sets it as the defining factor for cognitive ability.

## Multilayered sociograms provide insight into the evolution of children's peer preferences

Our exploration of sociograms considers the answers of children to five direct questions: preferred workmates, non-preferred workmates, preferred playmates, non-preferred playmates, and identified friends. With this information, we constructed a multilayer network with five layers, each representing one of these different aspects of the relationship among students.

First, we investigate the group structure of the different sociogram layers for each classroom using a stochastic block model (SBM) inferential approach (Methods and Supplementary Fig. S12). Interestingly, we find no systematic group structure across layers. In particular, for the working-related sociogram networks (those generated by the answers to "want to work with" and "don't want to work with" questions), we find that for the majority of classrooms (66%), all students are in a single group and that typically the structure of the sociogram can be explained by the number of connections of each student alone. This suggests that, when choosing academic partners, all students use the same "mechanism" and, as we show later, tend to prefer high-performing students.

However, the lack of a systematic group structure does not tell us anything about the similarities or differences between layers. To study these, we looked at the Jaccard index (that is, the edge overlap) between pairs of networks (Fig. 2 and Supplementary Fig. S13). We find that pairs of sociograms have more overlap than expected by chance, both for positive and negative answers, with observed assortativity being, on average, over 350% higher than the null model (all two-sided MWU tests have $p < 0.001$, and the average HLe >16 and lower 95% CI > 14). Importantly, however, we observe that the amount of overlap depends on age. Indeed, "friends" and "playing with" sociograms are increasingly similar with age. By contrast, "working with" and "playing with" sociograms for ages 6 and 7 have a high overlap, which then decreases with age (Fig. 2a).

The overall change from age 6 to age 15 shows a 28% decrease in overlap [$p < 0.001$ under a two-sided MWU test, and HLe = 3.9 (95% CI {3.2–4.6})], indicating strong age-related differences. However, the change in overlap varies from primary to secondary school. In primary school (ages 6–11), we observe a 13.3% decrease [$p < 0.001$, HLe = 2.0 (95% CI {1.4–2.8})], while in secondary school (ages 12–15), we see a 25.1% decrease [$p < 0.001$, HLe = 3.9]. The steeper decline in secondary school suggests that differentiation between work and play preferences is more pronounced during adolescence. The magnitude of these changes is substantial, with effect sizes well above conventional thresholds for large effects (HLe >1 and lower 95% CI >0.8). These differences are robust across different age comparisons, with particularly strong effects observed in the transitions to ages 14 and 15, where the decrease in overlap compared to earlier ages consistently shows large effect sizes [HLe's >3.0 (lower 95% CI >2.5)] and highly significant differences [all MWU $p$ values <0.001].

This suggests that, at older ages, children's academic awareness grows and decouples leisure and working preferences. Note that the peak in the Jaccard index at age 12 can be explained by the restart in the process of establishing work and friendship relationships with peers, which can associated with the transition from primary to secondary school in the Catalan system.

By contrast, the overlap between "not working with" and "not playing with" networks remains relatively stable throughout school years (Fig. 2b). This stability suggests that, unlike positive relationships, which is more differentiated with age, children's negative preferences persist strongly across both academic and social contexts. Once a child is rejected in one domain, they tend to be rejected in the other as well, and this pattern shows remarkable persistence across all ages. This could indicate that certain peer characteristics trigger a strong and lasting aversion that transcends specific contexts.

## Cognitive profiles and social interactions are interdependent

The analysis of sociograms reveals that work and play/friendship sociograms are not equivalent, especially at older ages, which suggests that students learn to discriminate between those individual attributes that make students better work partners. Our hypothesis is that there are changes in the relationship between the cognitive dimension of students and the structure of the sociograms that can partly explain differences in the work and play sociograms, and their age-related differences.

To test our hypothesis, we first investigate whether sociogram layers are always similarly assortative, that is, whether connected students in each of the different layers tend to share attributes. Indeed, groups of social interaction of children and adolescents in school environments have been found to be assortative in different dimensions including gender, ethnicity and academic achievement[55–57]. However, each study typically considers only a few school grades, so that the evolution of assortativity as students grow up has not been assessed.

Here, we look at assortativity in the gender and cognitive profile dimensions across different layers of interaction. Our analysis reveals strong gender homophily, with approximately 70−75% of connections being same-gender in early ages (6–11) for positive relationships ("working with", "playing with", and "friendship"). This tendency decreases during adolescence (age 12 onward), dropping to around 60−65% same-gender connections, consistent with the known onset of cross-gender relationships at that age[58]. By contrast, negative relationships are disassortative and is more disassortative with age (Supplementary Fig. S14), suggesting that even though cross-gender positive relationships are more common, conflicts remain predominantly cross-gender. Importantly, we see no clear differences in these gender patterns across sociograms that could explain the increasing differentiation between working and playing sociograms we observe in Fig. 2.

In the cognitive dimension, our analysis shows a similar picture. All positive relationships ("working with", "playing with", and "friendship") are, in general, assortative (Fig. 3); that is, students tend to want to interact with students with similar cognitive profiles for all measured dimensions. By contrast, negative relationships ("not working with" and "not playing with") have a tendency to be disassortative, that is, students tend to select students whose cognitive profile is less similar to their own than if they selected peers at random. Nevertheless, in this cognitive dimension, the story is more nuanced, since, by the end of secondary school, the choices of "not playing with" is neutral, indicating that children eventually stop taking into account the cognitive profile when rejecting playing mates.

To further assess the role of the cognitive profiles of students in the sociogram structure, we investigate the relationship between the position of students within the sociogram and their cognitive profile. Prior studies have shown that, especially during adolescence, a popularity hierarchy emerges in the social relationships among students that sets the direction of influence between peers[31,32]. Since social groups can have an influence on school engagement and the academic achievement of adolescent students[24], it is important to understand the relationship between cognitive dimension and sociogram status. To that end, we compute the social status of a student $s$ within sociogram $l$, via the PageRank index, $PR_{sl}$. To assess the relationship between status and cognitive profile, we obtain the Spearman's rank correlation between PageRank indices and the membership $\theta_{sg}$ of student $s$ to cognitive achievement group $g \in$ {high, low} obtained using the MMSBM approach described previously (Fig. 2). We then obtain averages for all classrooms within the same age and compare them to an expected baseline correlation (Fig. 4).

We find that there is a significant positive correlation between the status within the "working with" and the student's membership to the high-performing group (Fig. 4a) and a significantly negative correlation with the students' membership to the low-achievement group (Fig. 4b). As one may expect, we find opposite correlations for "not working with" relationships

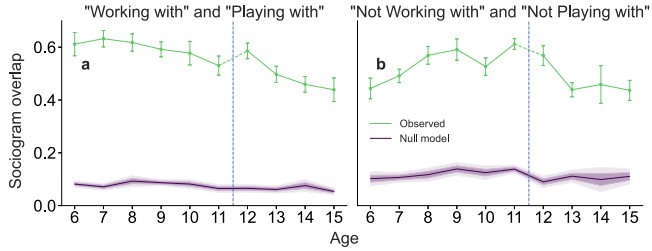

**Fig. 2 | Evolution of the overlap between work and play sociogram networks.**
Evolution of the overlap between pairs of sociograms corresponding to the following questions: We show the evolution of the overlap between pairs of sociograms corresponding to: **a** "playing with" and "working with"; **b** "not playing with", and "not working with". For each classroom, we measure the overlap between pairs of sociograms using the Jaccard index. Green lines show means across classrooms for each age, with error bars indicating the standard error of the mean. Sample sizes (number of independent classrooms) per age: $n = 18$ (age 6), $n = 26$ (age 7), $n = 19$ (age 8), $n = 20$ (age 9), $n = 21$ (age 10), $n = 17$ (age 11), $n = 25$ (age 12), $n = 10$ (age 13), $n = 13$ (age 14), $n = 13$ (age 15). Purple lines show the expected overlap from a null model that randomizes edges while preserving degree distribution, with shaded areas showing one and two standard deviations confidence intervals. The dashed vertical line marks the transition from elementary to secondary school.

(Fig. 4c-d). Interestingly, these correlations start out small and increase steadily until age 9, where they remain approximately constant throughout the following ages.

By contrast, correlations involving "playing with" and friendship relationships display a different pattern (Fig. 4e–j): While correlations between status and achievement groups are similar for ages 6 and 7, they are consistently less pronounced and ultimately compatible with the random expectation from age 12 onward, that is, during adolescence.

Our results thus show that, at older ages, significant differences between playing and working sociograms emerge and that these differences are related to the difference in status of students in the high-achievement group. Our results thus suggest that children gain awareness of their peers during the first two or three years of school, and that at that point start to tailor their social relationships to the type of activity, leading to the observed decrease in the overlap between working and leisure ties.

## Discussion

Using a large dataset involving over 5000 students aged 6 through 15, we have investigated the relationship between the cognitive abilities of the students and their social relationships within a scholar environment. The contributions of our study are two-fold. From the methodological data analysis point of view, our study investigates simultaneously the relationship between the cognitive achievement of students and the relationships of students in multiple social dimensions from childhood to adolescence, which include separate information about the choices of students for "working with", "not working with", "playing with", "not playing with", and "friends with".

Furthermore, our analysis provides several insights with respect to the cognitive profiles of students, the changes in the relationship between different sociograms with age, and how these changes are related to the position of students in the sociograms.

By using a network inference approach to model the achievement of students in the different cognitive tasks, we identify three underlying achievement groups; students in the same group perform similarly across all tasks (low, average, and high achievement). However, students do not belong to a single group but to mixtures of groups, so that we can characterize each students' achievement across all tasks with a cognitive profile vector of memberships to each achievement group. We find that students' cognitive profiles are typically mixtures of low and average, or average and high achievement. This finding aligns with hierarchical models of cognitive ability like the Cattell–Horn–Carroll theory that, while recognizing distinct cognitive abilities, consistently finds these abilities to be positively correlated[8,59].

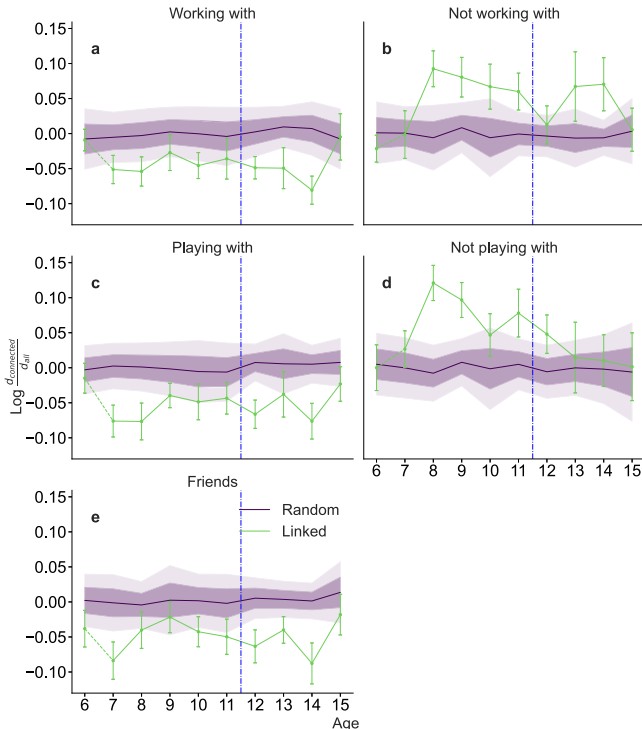

**Fig. 3 | Evolution of cognitive profile assortativity in sociograms throughout age.**
For each class group and sociogram layer, we compute the average distance between the cognitive profiles of students that are connected in that layer; we average this quantity over class groups in each age to obtain $d_{connected}$. We then compute the average cognitive distance between all possible pairs of students within a class, and the average over classes of the same age $d_{all}$. We show $\log\left[d_{connected}/d_{all}\right]$ (green line) for each age for the different layers. Sample sizes (number of independent classrooms analyzed) per age: $n = 18$ (age 6), $n = 26$ (age 7), $n = 19$ (age 8), $n = 20$ (age 9), $n = 21$ (age 10), $n = 17$ (age 11), $n = 25$ (age 12), $n = 10$ (age 13), $n = 13$ (age 14), $n = 13$ (age 15). Negative log-ratios indicate assortative sociograms ($d_{connected} < d_{all}$); positive log-ratios indicate disassortative sociograms. The purple line shows the null expectation for the cognitive distance, which we obtain by shuffling the cognitive vectors of students in each class. Shaded regions show one and two standard deviations around the mean of the null expectation (dark and light-shaded regions, respectively). The dashed vertical line marks the transition from elementary to secondary school. In **a**, **b**, we find the cognitive distance between children that are linked by a positive or negative "working with" choice, respectively. In **c**, **d**, the distance between positive and negative "playing with" choice, and, in **e**, the cognitive distance between children who choose another one as their friend.

Additionally, we find that social relationships within class groups change over time—young children (ages 6 and 7) have very similar sociograms for "working with", "playing with" and friendship relationships, which is consistent with previous studies for young children[57]; however, these sociograms are more different at older ages. In contrast, the overlaps between 'not working with' and 'not playing with' do not have a marked dependency on age.

We surmise that the changes in positive-relationship sociograms are due to the fact that children are able to, early on (and not only when they enter adolescence), distinguish between those people with whom they want to play and be friends and those students who are good working partners. This early differentiation aligns with broader developmental transitions occurring during this period, particularly what has been termed the "7 year shift" or entry into the "age of reason"[60]. During this period, children develop more sophisticated ways of understanding themselves and others, including more nuanced evaluations of competence across different domains. The timing of this differentiation in social preferences is particularly noteworthy, as it coincides with documented changes in how children conceptualize and evaluate intellectual competence[33]. While younger children tend to have a

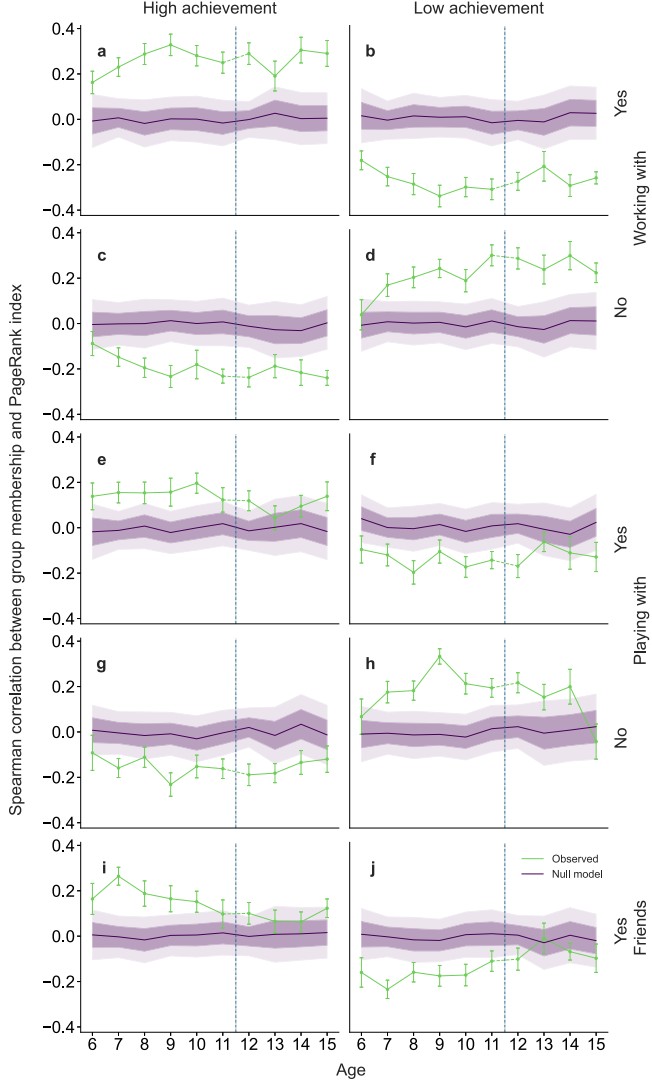

**Fig. 4 | Evolution of the association between cognitive profile and status in the sociograms.** We show Spearman's correlation between memberships to achievement groups $g \in \{high, low\}$, and PageRank indices (PR) within a sociogram $l$, as a function of age. **a** $g$ = high, $l$ = "working with"; **b** $g$ = low, $l$ = "working with"; **c** $g$ = high, $l$ = "not working with"; **d** $g$ = low, $l$ = "not working with"; **e** $g$ = high, $l$ = "playing with"; **f** $g$ = low, $l$ = "playing with"; **g** $g$ = high, $l$ = "not playing with"; **h** $g$ = low, $l$ = "not playing with"; **i** $g$ = high, $l$ = "friends with"; **j** $g$ = low, $l$ = "friends with". Green lines show the averages of correlations for class groups within an academic age. Blue lines show null expectation correlations computed by shuffling membership vectors of students. Dark and light-shaded areas correspond to one and two standard deviations around the mean null expectation, respectively. The dashed vertical line marks the transition from elementary to secondary school. The same number of classrooms have been analyzed for this plot: $n$ = 18 (age 6), $n$ = 26 (age 7), $n$ = 19 (age 8), $n$ = 20 (age 9), $n$ = 21 (age 10), $n$ = 17 (age 11), $n$ = 25 (age 12), $n$ = 10 (age 13), $n$ = 13 (age 14), $n$ = 13 (age 15).

more inclusive view of competence that combines social behavior, work habits, and academic achievement, older children develop a more differentiated understanding that allows them to distinguish between academic and social domains.

Indeed, our subsequent analysis confirms our assumption. First, we find no evidence for differences in gender assortativity across sociograms that can explain the differences in the sociogram structure we observe. Positive-relationship sociograms are all gender assortative. However, during adolescence, cross-gender interactions emerge, so that gender homophily is not as strong in secondary school. Instead, negative-relationship sociograms

are increasingly disassortative with age. For positive relationships, our results are consistent with previous studies about the role of gender within-school relationships[24,56,57] and also quantify the increase of cross-gender relationships among adolescents[58] in all social dimensions.

We find a similar pattern in terms of the assortativity of cognitive profiles, where all positive sociograms are assortative, and negative sociograms tend to be disassortative. The exception to this rule is the non-assortativity for the 'not playing with' sociogram in high school, which points to other attributes affecting children's choices at this stage.

Furthermore, the analysis of the relationship between the position of students within the sociograms and their cognitive profile vectors reveals clear differences in the positions that high-achieving students occupy. In "working with" sociograms, the membership to the high-achievement group is positively correlated with the status within the sociogram; and this correlation increases during the early school years and is the same for older students.

By contrast, in "playing with" and friendship sociograms, we find that this correlation decreases with age and is non-significant when students enter into adolescence. For "not working with" relationships, the results are the opposite, so they are negatively correlated with the status of high achievers and positively correlated with the status of low achievers. However, for "not playing with" relationships, the positive correlation with the status of low achievers is still substantial until age 14, which suggests there are some other factors that enhance dislikeness.

Our results indicate that not all relationships within a school environment are equivalent. All positive relations show cognitive assortativity, but the "working with" relationships show an additional competing effect: a general preference for high-performing students that increases with age. Specifically, while students tend to choose cognitively similar peers across all positive relationships, for academic work, they also display a strong hierarchical preference—students from all cognitive profiles are increasingly likely to select high-achieving peers as working partners as they advance through age. This hierarchical selection pattern aligns with the rich-get-richer popularity mechanism that is a main driver of social relationships during adolescence[24,32]. By contrast, the opposite effect is observed for low-achieving students, who are less often chosen to work with, and more often chosen to "not work with", a tendency that is maintained throughout all ages except, perhaps, for the oldest children (15 years).

The age-related differences we observe have important implications for understanding children's cognitive and social development. The emergence of differentiation between work and play relationships around age 7–8 aligns with fundamental changes in how children form the concept of "ability" in themselves and others[33] and process and use social comparison information[34]. Before this age, children's evaluations and social choices appear much more simplistic, and only at incorporating information about perceived academic ability on the second half of elementary school. Recent research has shown that during this same period, children begin developing increasingly differentiated self-concepts of their abilities across domains, particularly between mathematical and verbal domains[61]. Our findings reveal a complex relationship between these developments: while cognitive abilities remain largely unidimensional across domains (supporting the existence of a general factor), children develop distinct preferences for work versus play partners. This suggests that the development of differentiated academic self-concepts and social preferences may be more related to children's growing sophistication in social understanding than to actual differences in domain-specific cognitive abilities.

Our findings highlight several practical considerations for educational settings. Children begin differentiating between work and play preferences early in their school years, a pattern that teachers can anticipate and consider when designing classroom activities. These natural social dynamics provide valuable insight into how children organize their relationships throughout their development, though we note that we have not directly tested specific interventions. On the other hand, our results show that research associating peer relationships to academic outcomes should consider work and

friendship networks as distinct entities to better understand their contributions to student development.

On the other hand, the "not playing with" sociogram in high school students paints a different picture. Even though there is little to no effect of achievement on the "playing with" and friendship choices, the connection between cognitive profile and "not playing with" relationships remains strong until age 14, specially in the low achievement case. Furthermore, we have seen that the "not playing with" sociogram in high school is the only one where we find no cognitive assortativity. This suggests that some low achieving children tend to be rejected, even by other low achievers, a mechanism that is not present in other social interactions. This is compatible with these children being rejected due to aggressiveness[62] or anxious and withdrawn behaviors[63].

## Limitations

A limitation of our analysis that merits further study is that, although we have data from ages 6 to 15, we do not have long-term longitudinal data. The availability of longitudinal research on the relationships between cognitive and achievement profiles and social status on multilayered networks would be beneficial to further understand children's development in their educational context and, in particular, to establish causal relationships between academic achievement and social interactions.

## Data availability

All data needed for reproducibility can also be found in the previous repository. In the "data" directory you will find the original data, and, in the "output" directory, the cognitive data analyzed by the MMSBM algorithm that was later used for comparisons and plots.

Data and code can also be found at https://github.com/eudald-seeslab/stochastic-cognitive-blocks and in ref. 64.

## Code availability

The code for the analysis of this paper can be found at: https://github.com/eudald-seeslab/stochastic-cognitive-blocks, which uses the Python library "mmsbm", developed specifically for this project, which can be found at https://github.com/eudald-seeslab/mmsbm[65].

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

## Acknowledgements

This research was funded by the Spanish Government Ministerio de Ciencia e Innovación/AEI/10.13039 / 501100011033 (Project No. PID2022-142600NB-I00) and by the Government of Catalonia (Project No. 2021SGR-00170 and Industrial Doctorate No. DI-128). We thank Aleix Saló-Braut for his contribution to the conceptualization and creation of the visual abstract. The funders had no role in study design, data collection and analysis, the decision to publish, or preparation of the manuscript.

## Author contributions

Eudald Correig-Fraga wrote the code and run all computational experiments. Eudald Correig-Fraga, Roger Guimerà, and Marta Sales-Pardo designed the research, analyzed the data, and wrote the paper.

## Competing interests

The authors declare no competing interests.
