## [Transparent Peer Review file · Communications Psychology]

Interplay between children's cognitive profiles and within-school social interactions is nuanced and differs across ages

Corresponding Author: Mr Eudald Correig-Fraga

Version 0:

Decision Letter:

Dear Mr Correig-Fraga,

Thank you for your patience during the peer-review process. Your manuscript titled "Large-scale analysis of the evolving interplay between children's cognitive profiles and social interactions during school years" has now been seen by 2 reviewers, and I include their comments at the end of this message. They find your work of interest but raised some important points. We are interested in the possibility of publishing your study in Communications Psychology, but would like to consider your responses to these concerns and assess a revised manuscript before we make a final decision on publication.

We therefore invite you to revise and resubmit your manuscript, along with a point-by-point response to the reviewers. Please highlight all changes in the manuscript text file.

Editorially, we consider the concerns regarding the robustness and the size of the reported effects critical and ask that the revised manuscript follows the journal's statistical reporting guidelines including supporting all inferential statements with fully reported statistics which includes reporting effect sizes and their confidence intervals. Additionally, a revision should include evaluation of the robustness of the results and provide a deeper and critical interpretation of the findings and their impact.

I am attaching an Editorial Requests Table that details critical reporting requirements for the revised manuscript. Please attend to each item and ensure your manuscript is fully compliant. If your revised manuscript is not aligned with these requests on major issues, such as those concerning statistics, it may be returned to you for further revisions without re-review.

Please submit the following items:

- Revised manuscript
- Point-by-point response to the referees' comments
- Cover letter (as a separate document)
- [Nature Research Reporting Summary](https://www.nature.com/documents/nr-reporting-summary.zip)
- [Editorial Policy Checklist](https://www.nature.com/documents/nr-editorial-policy-checklist.pdf)
- Completed Editorial Request Table (attached).

via this link: Link Redacted .

Additional guidance is available in our style and formatting guide Communications Psychology formatting guide.

Best regards,

Yana Fandakova

Yana Fandakova
External Editor
Communications Psychology

REVIEWER EXPERTISE:

Reviewer #1 mixed membership stochastic block models, peer relationships / social networks, adolescent (cognitive & social) development

Reviewer #2 peer relationships / social networks, adolescent (cognitive & social) development

REVIEWER REPORTS:

Reviewer #1 (Remarks to the Author):

The authors use sociometric analyses to study how children's perceptions of whom they want to work with and play with vary across ages, particularly as they relate to children's academic achievement. This is an interesting paper. The data and some of the findings are pretty interesting - I could imagine this paper being published in this journal - but the theoretical background and interpretation of findings are quite shallow. I have a major concern and some less major ones, the latter of which I'm confident can be easily addressed.

My strongest concern is about the size and robustness of the main findings - that there is differentiation in whom kids want to play with from whom they want to work with during these years (Fig. 3). I don't see any inferential statistics nor a clear discussion of the size of this effect. Relative to the null model, I don't see a strong developmental trend in Fig. 3. What should we make of this? Relatedly, I would like to see the magnitudes of all effects discussed, including those related to gender-specific preferences across age and the cognitive profile assortativity, in a precise and thoughtful way. How large, how robust, are these effects? I realize the measures of effects may be unfamiliar to many readers, but perhaps the authors could use terms like "a 5% decrease in assortativity on X relative to random sorting from grade X to Y" or something like that.

If robust, the main finding is interesting, but does it really imply that:

"promoting diverse interactions and collaborations may encourage broader academic achievement and social development"? I like this idea, but I'm not sure it's true; first, it may be that unique tastes for friends are largely endogenous to preexisting "broader academic achievement and social development"; second, although it is a neat idea that a broader peer group will result in broader positive developmental outcomes, it would be nice to see whether this is true relative to, say, a very narrow peer group composed of individuals with broad academic achievement and social development. In the latter case, it may still be *socially* optimal for people to have diverse friend groups, but the issue that it may not be *individually* optimal would be an important challenge worth worrying about.

For these reasons, I am concerned the framing of the implications of the work is overly simplistic and optimistic. The paper is interesting and some version of it should be published somewhere with these implications discussed a bit more deeply or merely removed.

In the framing, the authors should at least address previous findings around the differentiation of children's interests, identities, and abilities across this developmental period. Off the top of my head, here's a paper with a relevant finding and many older relevant citations in the intro:

Wan, S., Lauermann, F., Bailey, D. H., & Eccles, J. S. (2021). When do students begin to think that one has to be either a "math person" or a "language person"? A meta-analytic review. *Psychological Bulletin*, 147, 867–889.

"that is, that in statistical terms there are no students that perform very well in one type of task and badly in others"

I don't think I agree with this interpretation of this result. Rather, the assumption that the latent group structure that fit the data best is that there are no children with true scores that vary substantially across tests. This may in some cases be a useful assumption: the positive manifold is a 100 year old finding in psychology after all. But it is not literally true that there are no students that do well on one task and poorly on others, and I highly doubt it's true in these data (if it is, it's a major indictment of the measures).

Reviewer #2 (Remarks to the Author):

The paper meets the standards of the journal, presenting original research that contributes to the literature in this field. Its major claims, focused on the interactions of children and adolescents with their peers, differentiating between different layers (work and leisure), are likely to be of interest to the community. Additionally, the MMSBM approach is well justified and is clearly proven to be more powerful in modeling overlapping layers and complex relationships. The conclusions are convincing, and the study's insights have the potential to influence thinking within the field. Nevertheless, I will suggest some comments that can be addressed to improve it.

In the abstract, it would be beneficial to specify that the sample consists of Spanish students, as this contextual detail provides important clarity for readers. Additionally, it might be more effective to refer to the participants by their years instead of grades, as this could facilitate a better understanding and identification of the children, particularly for an international audience.

This comment can also be applied to the introduction since the first mention of the sample's origin is implied from the Spanish educational system in line 127, which may seem quite out of the blue. Including this detail earlier would improve clarity and coherence.

In line 62, the phrase "in the spirit of 31" should include the actual authors' names instead of a numerical reference.

Just as a personal concern, I wonder if there are any studies that support classifying the emotion recognition task in group 1 along with the attention-related tasks. Adding such references could provide extra context and further support to justify this classification.

It would be worth considering whether the conclusion (line 96) about the existence of a single axis of performance across tasks is consistent with the literature's findings.

The conclusion in line 130 can be confusing (The lack of change is due to the immediate recognition of negative peer characteristics), it could be rewritten.

Panel d of Figure 2 might be sent to the appendix since it only illustrates the choice of method.

Since you perform an assortativity analysis by gender, it would be interesting to know if the previous analyses were also conducted by gender. Did you find the same profiles, or are there any significant differences that could be included in the study? This additional exploration might provide further depth to the findings.

Also, it might also be interesting to mention gender homophily. Including this perspective could provide additional insights into individual preferences for same-gender connections, complementing the broader analysis of gender assortativity.

The explanation of the assortative analysis across the different layers is not entirely clear to me. Initially, it seems that all students, regardless of their cognitive profile, prefer to choose peers with high cognitive abilities for work interactions. However, the conclusion that positive relationships are assortative suggests that students with high cognitive abilities tend to interact with others with similarly high abilities, while those with low abilities interact with peers of similarly low abilities. I understand that assortativity does not only focus on a single characteristic of the individuals but what would be the practical statement that follows from the authors' results?

The results regarding students' positions, specifically their popularity within the sociograms, are not entirely clear. Popularity is mentioned only briefly in line 160, and this aspect is not anticipated or contextualized in the introduction.

The authors write "Our findings are of especial practical relevance because they may offer possible ways to help adolescent students improve academic performance" this conclusion in line 236, appears overly generalized and not sufficiently supported by the results obtained through the analyses presented in the study. It would be beneficial to clearly connect this conclusion to the specific findings and data from the study to provide a more grounded and robust argument.

Finally, the indirect questions in the sociogram part are very interesting and I wonder if they are in your future research perspective to extend the insights of this study.

Version 1:

Decision Letter:

Dear Mr Correig-Fraga,

Your manuscript titled "Large-scale analysis of the evolving interplay between children's cognitive profiles and social interactions during school years" has now been seen by our reviewers, whose comments appear below. In light of their advice I am delighted to say that we are happy, in principle, to publish a suitably revised version in Communications Psychology.

We therefore invite you to revise your paper one last time to address the remaining concerns of our reviewers and a list of editorial requests. At the same time we ask that you edit your manuscript to comply with our format requirements and to maximise the accessibility and therefore the impact of your work.

EDITORIAL REQUESTS:

SUBMISSION INFORMATION:

OPEN ACCESS:

At acceptance, you will be provided with instructions for completing the open access licence agreement on behalf of all authors. This grants us the necessary permissions to publish your paper. Additionally, you will be asked to declare that all required third party permissions have been obtained, and to provide billing information in order to pay the article-processing

charge (APC).

* **DATA AVAILABILITY:**

Link Redacted

Best regards,

Jennifer Bellingtier

Jennifer Bellingtier, PhD
Senior Editor
Communications Psychology

Yana Fandakova
External Editor
Communications Psychology

REVIEWER EXPERTISE:

Reviewer #1 mixed membership stochastic block models, peer relationships / social networks, adolescent (cognitive & social) development

Reviewer #2 peer relationships / social networks, adolescent (cognitive & social) development

REVIEWERS' COMMENTS:

Reviewer #1 (Remarks to the Author):

I thank the authors for their thorough response to my initial review.

Reviewer #2 (Remarks to the Author): no further comments to the Author.

Reply to Reviewer #1

COMMENT: The authors use sociometric analyses to study how children's perceptions of whom they want to work with and play with vary across ages, particularly as they relate to children's academic achievement. This is an interesting paper. The data and some of the findings are pretty interesting - I could imagine this paper being published in this journal - but the theoretical background and interpretation of findings are quite shallow. I have a major concern and some less major ones, the latter of which I'm confident can be easily addressed.

Thank you for your positive assessment of our work and for the constructive feedback. We have carefully addressed your concerns about the theoretical background and interpretation of findings in our responses below and in the revised manuscript.

COMMENT: My strongest concern is about the size and robustness of the main findings - that there is differentiation in whom kids want to play with from whom they want to work with during these years (Fig. 3). I don't see any inferential statistics nor a clear discussion of the size of this effect. Relative to the null model, I don't see a strong developmental trend in Fig. 3. What should we make of this? Relatedly, I would like to see the magnitudes of all effects discussed, including those related to gender-specific preferences across age and the cognitive profile assortativity, in a precise and thoughtful way. How large, how robust, are these effects? I realize the measures of effects may be unfamiliar to many readers, but perhaps the authors could use terms like "a 5% decrease in assortativity on X relative to random sorting from grade X to Y" or something like that.

We have looked at the statistical significance, robustness and effect size of our comparison of 'working with' and 'playing with' relationships.

First, we have assessed the statistical significance of the difference in sociogram overlap with respect to the null model. We find that 'working with' and 'playing with' sociograms have more overlap than expected by chance, both for positive and negative answers, with the overlap being on average over 350% higher than the null model (all $p < .001$, average Cohen's $d > 15$).

Second, to address the concern about robustness, we have added pairwise comparisons between grades, which show that these differences are consistent across different grade comparisons. The overall change from grade 1 to grade 10 shows a 28% decrease in overlap (t-test p -value $< .001$, Cohen's $d = -3.89$). In fact, this change isn't linear but shows distinct patterns in primary and secondary school. In primary school (grades 1-6), we observe a 13.3% decrease ($p < .001$, $d = -2.02$), while in secondary school (grades 7-10), we see a 25.1% decrease ($p < .001$, $d = -3.9$). Therefore we can conclude not only that the changes in overlap across sociograms with age are statistically significant, but also, with effect sizes well above conventional thresholds for large effects ($|d| > 0.8$), and robust across different grade comparisons.

CHANGES: We have now added a comprehensive statistical analysis of the trends of sociogram overlap and effect sizes in the Results section. Furthermore, we have added hypothesis testing results and effect sizes where appropriate, as well as standard error bars in the appropriate plots to visually convey the significance of changes in overlap values.

COMMENT: If robust, the main finding is interesting, but does it really imply that: "promoting diverse interactions and collaborations may encourage broader academic achievement and social development"? I like this idea, but I'm not sure it's true; first, it may be that unique tastes for friends are largely endogenous to preexisting "broader academic achievement and social development"; second, although it is a neat idea that a broader peer group will result in broader positive developmental outcomes, it would be nice to see whether this is true relative to, say, a very narrow peer group composed of individuals with broad academic achievement and social development. In the latter case, it may still be *socially* optimal for people to have diverse friend groups, but the issue that it may not be *individually* optimal would be an important challenge worth worrying about.

For these reasons, I am concerned the framing of the implications of the work is overly simplistic and optimistic. The paper is interesting and some version of it should be published somewhere with these implications discussed a bit more deeply or merely removed.

We agree with you that some conclusions could be considered far fetched. We have made substantial changes to the manuscript by removing sentences from the abstract and discussion and substituting them with more cautious statements.

COMMENT: In the framing, the authors should at least address previous findings around the differentiation of children's interests, identities, and abilities across this developmental period. Off the top of my head, here's a paper with a relevant finding and many older relevant citations in the intro: Wan, S., Lauermaun, F, Bailey, D. H., & Eccles, J. S. (2021). When do students begin to think that one has to be either a "math person" or a "language person"? A meta-analytic review. *Psychological Bulletin*, 147, 867-889.

Thank you for pointing out this part of the literature we were not familiar with. We have updated the introduction and the conclusions with references to the work of Wan *et al* [6] as well as other very relevant works cited therein:

- We have incorporated citations to Stipek *et al.* [5] and Ruble *et al.*[4] to the introduction to better contextualize the developmental changes that students undergo in the beginning of compulsory education. Specifically, according to these studies at 6-7 years of age children learn to differentiate [intellectual] ability from other good traits, and start comparing their abilities with those of others.
- We have incorporated citations to the work of Stipek *et al.* [5] and Davis-Kean *et al.* [2], to introduce the concept of entering "age of reason" in children of ages 5-7, which aligns very well with our findings and our interpretation of children learning to separate work from leisure at that age.

COMMENT: "that is, that in statistical terms there are no students that perform very well in one type of task and badly in others" I don't think I agree with this interpretation of this result. Rather, the assumption that the latent group structure that fit the data best is that there are no children with true scores that vary substantially across tests. This may in some cases be a useful assumption: the positive manifold is a 100 year old finding in psychology after all. But it is not literally true that there are no students that do well on one task and

poorly on others, and I highly doubt it's true in these data (if it is, it's a major indictment of the measures).

What you say is correct. Our results show that, the best latent space is one in which students have similar achievement across tasks. This is indeed the case for the majority of the students we analyze. However, we find that about 35% of students present extreme results that differ from their average trend. For instance, students in the high-achieving profile can have scores in the lowest quintile in some tasks (Fig. R1A) and, conversely, students in the low-achieving group can have scores in the highest quintile for specific tasks (Fig. R1B) or average students who score in highest quintile in some tasks and in the lowest in others (Fig. R1C). However, 92% of the students presenting extreme results only do so in at most 2 tasks (Fig. R1). Therefore, ours is not only the best latent model of all the models we considered, but also provides an accurate description of scoring patterns of students.

CHANGES: We have modified the text to be more precise and to incorporate this more nuanced analysis of students' achievement. We have also added Fig. R1 tot he supplementary material.

Figure R1: Distribution of the number of tasks with extreme scores for different groups of students. **A.** For students in the high-achieving group (37% of the students' pool), we show the relative percentage of students that score in the lowest quintile in n tasks. **B.** For students in the low-achieving group (22% of the pool), we show the relative percentage that score within the highest quintile in n tasks. **C.** For average-achieving students (41%), we show a heat map with the relative percentage of students who have x tasks with scores in the top quintile (5) and y tasks with scores in the bottom quintile (1). In **A**, we can see that 58.7% of high-achieving students have no tasks at the lowest percentile (compared to the 0% for low-achieving and the 31.5% for the average-achieving students). In **B**, we can see that 62.8% of the low-achieving students have no tasks in the highest quintile, compared to the 1.1% of the high performers and the 44.6% for the average achieving students. In **C**, we see that the majority of average students have some tendency to either score low or high without mixing, and that only 30% have both extreme scores in two or more tasks.

Finally, please note that we have moved the "Methods" section after the Introduction in compliance with the journal requisites.

Reply to Reviewer #2

COMMENT: The paper meets the standards of the journal, presenting original research that contributes to the literature in this field. Its major claims, focused on the interactions of children and adolescents with their peers, differentiating between different layers (work and leisure), are likely to be of interest to the community. Additionally, the MMSBM approach is well justified and is clearly proven to be more powerful in modeling overlapping layers and complex relationships. The conclusions are convincing, and the study's insights have the potential to influence thinking within the field. Nevertheless, I will suggest some comments that can be addressed to improve it.

Thank you for the positive assessment of our work and for the comments which have allowed us to improve our manuscript.

COMMENT: In the abstract, it would be beneficial to specify that the sample consists of Spanish students, as this contextual detail provides important clarity for readers. Additionally, it might be more effective to refer to the participants by their years instead of grades, as this could facilitate a better understanding and identification of the children, particularly for an international audience. This comment can also be applied to the introduction since the first mention of the sample's origin is implied from the Spanish educational system in line 127, which may seem quite out of the blue. Including this detail earlier would improve clarity and coherence.

All the students were from Catalan schools and, therefore, within the Catalan education system as we now mention early on in the paper. Note that, in Spain, the educational competences are transferred to the local governments, so that different autonomous communities have different education systems in agreement with the general educational program established by the central government. For instance, in Catalonia students learn Spanish, Catalan and a foreign language, have a different organization of secondary education programs, and different exams to grant entry to university level education.

COMMENT: In line 62, the phrase "in the spirit of 31" should include the actual authors' names instead of a numerical reference.

We have included an explicit mention to the authors of the paper (Gerlach *et al.*).

COMMENT: Just as a personal concern, I wonder if there are any studies that support classifying the emotion recognition task in group 1 along with the attention-related tasks. Adding such references could provide extra context and further support to justify this classification.

We could not find any literature in this direction. This the reason why, in our case, we hypothesize that the achievement of students in this task majorly reflects the fact that the task is at the very end of a 1 hour test that requires student's continued concentration and less on their specific abilities to identify emotions.

COMMENT: It would be worth considering whether the conclusion (line 96) about the existence of a single axis of achievement across tasks is consistent with the literature's findings.

In the literature, there is ample background in agreement with our findings. In particular, the Cattell-Horn-Carroll (CHC) theory is a classical model of cognitive ability which says that while individuals might have distinct cognitive abilities, these abilities are consistently found to be positively correlated [3, 1].

CHANGES: We have added references to the CHC theory to provide a better contextualization of our findings.

COMMENT: **The conclusion in line 130 can be confusing (The lack of change is due to the immediate recognition of negative peer characteristics), it could be rewritten.**

We have rewritten the sentence for more clarity.

COMMENT: **Panel d of Figure 2 might be sent to the appendix since it only illustrates the choice of method.**

We have moved panel d to the appendix.

COMMENT: **Since you perform an assortativity analysis by gender, it would be interesting to know if the previous analyses were also conducted by gender. Did you find the same profiles, or are there any significant differences that could be included in the study? This additional exploration might provide further depth to the findings. Also, it might also be interesting to mention gender homophily. Including this perspective could provide additional insights into individual preferences for same-gender connections, complementing the broader analysis of gender assortativity.**

To compare the cognitive profiles of the two genders, we performed a two sided Mann Whitney U test of the membership values for each of the membership groups ("Low", "Average", and "High") per gender. We have not found any systematic gender differences in any of the variables of our study with the exception of the student group of low performers in which there is a larger fraction of boys than girls (with an effect size of 0,10 measured by Cohen's d, and $p < 0,001$) (see Fig. R2).

We now explicitly discuss homophily when discussing the results about gender assortativity patterns.

CHANGES: In the manuscript, we now explicitly discuss homophily and its possible relationship with our findings.

COMMENT: **The explanation of the assortative analysis across the different layers is not entirely clear to me. Initially, it seems that all students, regardless of their cognitive profile, prefer to choose peers with high cognitive abilities for work interactions. However, the conclusion that positive relationships are assortative suggests that students with high cognitive abilities tend to interact with others with similarly high abilities, while those with low abilities interact with peers of similarly low abilities. I understand that assortativity does not only focus on a single characteristic of the individuals, but what would be the practical statement that follows from the authors' results?**

We agree with you that that the relationship between cognitive assortativity and hierarchical preferences in the 'working with' sociogram can be confusing.

Our analysis reveals two concurrent effects:

Figure R2: Distribution of group memberships per gender. When comparing the membership to the low, average, and high membership groups using two-sided Mann Whitney U tests, we find that only the membership to the low group is significant, with an effect size of 0,10 measured by Cohen’s d, and $p < 0,001$.

- Cognitive assortativity (with respect to the null expectation) across all positive relationships (working, playing, friendship) where students tend to choose peers with similar cognitive profiles.
- A hierarchical effect in the ‘working with’ sociogram, where high-achieving students tend to hold more important positions (higher page rank) in the sociogram. This could be either because there is a preference to work with high-achieving students or with students that prefer to work with these high-achieving students, thus establishing a hierarchy in the working-with relationships with high-achieving students at the top. This is why we mention that the mechanism is similar to the popularity mechanism by which teenagers in general prefer to establish relationships with peers that they perceive as being more popular than themselves.

CHANGES: We have revised the relevant section to make this distinction clearer and to better explain how these two effects combine to shape students’ choices of working partners.

COMMENT: **The results regarding students’ positions, specifically their popularity within the sociograms, are not entirely clear. Popularity is mentioned only briefly in line 160, and this aspect is not anticipated or contextualized in the introduction.**

To provide better contextualization for our results, we now mention the concept of popularity in the introduction.

COMMENT: **The authors write “Our findings are of especial practical relevance because they may offer possible ways to help adolescent students improve academic achievement” this conclusion in line 236, appears overly generalized and not sufficiently supported by the**

results obtained through the analyses presented in the study. It would be beneficial to clearly connect this conclusion to the specific findings and data from the study to provide a more grounded and robust argument.

We agree with both reviewers on this point; we have been overly optimistic in our conclusions. We have removed strong statements, and moderated the conclusions we extract from our analysis.

COMMENT: Finally, the indirect questions in the sociogram part are very interesting and I wonder if they are in your future research perspective to extend the insights of this study.

We agree with you that indirect questions are of interest. In future work, we might look at the relationship between the answer to these indirect questions, cognitive profiles and the answers to direct sociogram questions. Of special interest might be to explore how the negative relationships interact with students perceived aggression in their peers, and which are the cognitive profiles of these perceived aggressive children.

Finally, please note that we have moved the “Methods” section after the Introduction in compliance with the journal requisites.

References

- [1] J. B. Carroll. *Human cognitive abilities: A survey of factor-analytic studies*. Human cognitive abilities: A survey of factor-analytic studies. Cambridge University Press, 1993. Pages: ix, 819.
- [2] P. E. Davis-Kean, L. R. Huesmann, J. Jager, W. A. Collins, J. E. Bates, and J. E. Lansford. Changes in the Relation of Self-Efficacy Beliefs and Behaviors Across Development. *Child Development*, 79(5):1257–1269, Sept. 2008.
- [3] K. S. McGrew. CHC theory and the human cognitive abilities project: Standing on the shoulders of the giants of psychometric intelligence research. *Intelligence*, 37(1):1–10, Jan. 2009.
- [4] D. N. Ruble, A. K. Boggiano, N. S. Feldman, and J. H. Loebel. Developmental analysis of the role of social comparison in self-evaluation. *Developmental Psychology*, 16(2):105–115, 1980. Place: US Publisher: American Psychological Association.
- [5] D. Stipek and D. M. Iver. Developmental Change in Children’s Assessment of Intellectual Competence. *Child Development*, 60(3):521, June 1989.
- [6] S. Wan, F. Lauermann, D. H. Bailey, and J. S. Eccles. When do students begin to think that one has to be either a “math person” or a “language person”? A meta-analytic review. *Psychological Bulletin*, 147(9):867–889, Sept. 2021.